# The comprehensive researcher development framework (CRDF): Core learning outcomes for research training

**Janet L. Branchaw**[1,2*], **Amanda R. Butz**[2], **Joseph C. Ayoob**[3]

**1** Department of Kinesiology, University of Wisconsin - Madison, Madison, Wisconsin, United States of America, **2** Wisconsin Institute of Science Education and Community Engagement, University of Wisconsin - Madison, Madison, Wisconsin, United States of America, **3** Department of Computational and Systems Biology, University of Pittsburgh, Pittsburgh, Pennsylvania, United States of America

* branchaw@wisc.edu

## Abstract

Becoming a researcher involves the iterative development of deep disciplinary knowledge, specific technical skills, and psychosocial attitudes, behaviors, and beliefs. Consequently, training researchers is resource- and time-intensive. In addition, expectations can be opaque because the traditional apprenticeship model used in research training is idiosyncratic, defined by norms and traditions that vary across disciplines. To align and make research training expectations more transparent, we developed the **Comprehensive Researcher Development Framework (CRDF)** by extracting and analyzing learning outcomes from 56 previously published evidence-based frameworks from across disciplines. The individual frameworks each addressed a limited range of training stages (e.g., undergraduate only), focused on a subset of learning outcomes (e.g., technical skills), and/or included a single or narrow subset of disciplines (e.g., biomedical sciences). The CRDF derived from these frameworks includes 79 core learning outcomes nested under 8 areas of researcher development that are supported by evidence of content validity collected from experts in the research community. The CRDF builds consensus across disciplines and addresses undergraduate through postdoctoral career stages to define a coherent continuum of research learning outcomes that can be used to monitor and study researcher development. The CRDF does not replace existing discipline-based or training stage specific frameworks but rather can link and coordinate their use. The CRDF can be used by research training program directors to design new or refine existing research training programs, track individual research mentee development over time, and demystify the research training process for mentors and mentees. The CRDF can also be used by scholars studying researcher development to link data on core learning outcomes across research training programs, stages, and disciplines.

**Data availability statement:** All relevant data are within the paper and its Supporting Information files.

**Funding:** The author(s) received no specific funding for this work.

**Competing interests:** The authors have declared that no competing interests exist.

## Introduction

Attracting motivated students with high potential from diverse backgrounds to research careers and providing them with rigorous, yet supportive research training experiences [1,2] is key to building a strong and innovative research workforce. However, training individuals to become researchers is complex and takes time. It involves the development of deep disciplinary knowledge, specific technical skills, and psychosocial attitudes, behaviors, and beliefs that promote integration and belonging in disciplinary research communities [3–6]. The diverse ways of knowing [7], research methodologies, and types of research projects across disciplines coupled with the apprentice model of training used in many disciplines produce research training journeys that are unique to each student. Consequently, designing and assessing the effectiveness of varied research training pathways can be challenging.

Though approaches and methods for conducting research are always evolving, and learning to do research is a lifelong process, *formal* research training in most disciplines begins during undergraduate education, is the primary focus of graduate education, and may be extended with postdoctoral training depending on the discipline. Ideally, formal research training across training stages forms a continuum that builds increasingly sophisticated disciplinary knowledge, perspectives, expertise, professional responsibilities, and relationships that are needed to successfully design and conduct rigorous research. However, consensus across disciplines and training stages about research learning outcomes is limited. Without defined common core learning outcomes, it is difficult to coordinate research training across programs, training stages, and between mentors, which can lead to contradicting or ill-defined expectations for mentees.

Inconsistencies across training programs and unclear expectations pose challenges for mentees. These challenges sometimes result in talented mentees abandoning a seemingly uncertain research career path for more well-defined, familiar, or lucrative opportunities outside of research [8–14]. This can be particularly important for mentees with limited research backgrounds, for whom the research culture is unfamiliar and persistence along a research career path uncertain. Systematizing research training and clarifying expectations is key to retaining the talented, high potential students we need to build the research workforce [3,4].

To clarify, align, and study research training, scholars (including two of the authors, [15]) have conducted studies to identify and understand how researchers develop and have published researcher development frameworks and/or assessments based on their findings (S1 Appendix). Conceptual frameworks are structures that describe "the factors and/or variables involved in (a) study and their relationships to one another" [16]. They can be used to guide training programs, mentors, and mentees in selecting and evaluating the impact of training activities, as well as to assess and monitor mentee development as a researcher over time. The individual frameworks in published studies, however, typically span a limited range of training stages (e.g., undergraduate only), focus on a subset of learning outcomes (e.g., technical skills), and/or include a single or narrow subset of disciplines (e.g., biomedical sciences).

Consequently, building a coherent continuum of research training or studying researcher development across training stages and/or disciplines is challenging.

We leveraged the prior work done on the discipline and training stage specific frameworks to demonstrate consensus across disciplines and develop the Comprehensive Researcher Development Framework (CRDF). We analyzed 56 individual frameworks to identify the common knowledge, skill, and psychosocial attitudes, behaviors, and beliefs that researchers develop from the undergraduate through postdoctoral training stages. Through multiple phases of input from the research community, we defined and confirmed the importance of 79 core learning outcomes and organized them into 8 areas of researcher development (Table 1).

Importantly, the CRDF is not meant to replace existing disciplinary or training stage specific frameworks but rather provide a comprehensive benchmark against which these frameworks can be compared and through which they can be linked. The CRDF can be used to study researcher development and research training programs across disciplines and training stages, to guide training program development when specific frameworks do not exist, or as a template from which to build new frameworks for specific disciplines or training stages. The CRDF can also be shared with mentees to make the expectations of research training explicit and empower them to take responsibility for their research training experience. Likewise, it can be shared with mentors to help them articulate expectations and assess their mentees' progress. Tools based on the CRDF to support these uses are provided in the S2a and S2b Appendix.

## Methods

An overview of the process used to develop the CRDF is presented in Fig 1. All human subjects research presented in this article was approved by the University of Wisconsin - Madison's Institutional Review Board, protocol # 2024−0876.

1. Synthesize research

We followed the five stage process of research synthesis outlined by Cooper [17]: Problem Formulation; Literature Search; Data Evaluation; Analysis & Interpretation; and Presentation of Results. The first four stages are described in the Methods section and the final stage in the Results section.

### Problem formulation

We sought to answer the question: *What common sets of knowledge, skills, and psychosocial attitudes, behaviors, and beliefs do researchers-in-training develop across different disciplines and training stages (i.e., undergraduate through postdoctoral)?*

### Literature search

We conducted a comprehensive review of the literature published in the last quarter century (2000–2024) to identify research development frameworks across disciplines from undergraduate to postdoctoral career stages that were supported with evidence of validity. Prior to conducting the literature search, key terms to inform the search were defined:

- **Research knowledge:** specific disciplinary knowledge; knowledge about the process of investigating or exploring the unknown needed to conduct disciplinary research and/or to advance the discipline; knowledge of the structures and resources that support research in the discipline (e.g., peer review practices, funding mechanisms).

- **Research skills:** sets of skills that trainees develop while engaging in research that advance their development as researchers.

- **Research psychosocial attitudes, behaviors, and beliefs:** disciplinary cultural norms, ways of networking, and engaging in interpersonal interactions; development of identity as a researcher.

**Table 1. Comprehensive Researcher Development Framework (CRDF).**

**1. Foundational Disciplinary Knowledge** *Understand historical and emerging disciplinary content, concepts, frameworks, and theories and how they relate to other disciplines*.
**Researchers:**

1.01 know the fundamental content in their discipline (e.g., frameworks, theories, and models).

1.02 know the history of knowledge generation in their discipline.

1.03 know the processes by which new knowledge is generated and evaluated.

1.04 ground hypotheses and research questions in established disciplinary knowledge, theories, frameworks, or observations.

1.05 know inferences and implications of research findings.

1.06 know the ways that content knowledge from other disciplines is related to content knowledge in their discipline.

**2. Practical and Cognitive Research Skills** *Know and apply disciplinary knowledge, technical, and reasoning skills to conduct research that advances knowledge in the discipline.*
**Researchers:**

2.01 use tools and databases to search the disciplinary literature.

2.02 use literature search strategies that identify relevant prior research.

2.03 use logical and critical thinking in evaluating research.

2.04 connect diverse research ideas and approaches in novel and creative ways.

2.05 consider alternative approaches and interpretations of research.

2.06 make connections between content knowledge in their discipline and content knowledge in other disciplines.

2.07 identify gaps in existing knowledge or research results to investigate.

2.08 set research goals.

2.09 use disciplinary theories, frameworks and models in designing research studies.

2.10 provide a logical rationale for their study designs.

2.11 know assumptions and limitations in study designs (e.g., reporting uncertainty/error).

2.12 formulate hypotheses and research questions that can be systematically tested or investigated.

2.13 select appropriate methods to investigate research questions.

2.14 follow standard protocols to collect and store research data.

2.15 have up-to-date technical skills to conduct research in the discipline.

2.16 develop new data collection or analytical methods when needed.

2.17 use troubleshooting skills to address theoretical or technical problems in research.

2.18 apply the appropriate analytic and statistical methods to analyze data.

2.19 use disciplinary theories, frameworks and models in analyzing data.

2.20 interpret the results of data analyses (e.g., coding, mathematical, and statistical calculations).

2.21 interpret or synthesize research findings.

2.22 propose new inferences and implications of research findings (their own and/or others').

2.23 draw conclusions from research findings.

2.24 refine existing and/or contribute new disciplinary theories, frameworks, and models based on research findings.

**3. Ethical and Responsible Research Practices** *Follow guidelines for responsible conduct of research and recognize and respond to ethical issues that impact and emerge from conducting research.*
**Researchers:**

3.01 follow research safety regulations.

3.02 follow disciplinary data ownership and stewardship practices.

3.03 follow ethical guidelines for working with research data.

3.04 follow guidelines for ethical treatment of research subjects (e.g., individuals, communities, animals, etc.).

3.05 follow guidelines for conducting rigorous and reproducible research in their discipline.

3.06 follow disciplinary norms and policies regarding credit for contributions to research (e.g., citing previous research, authorship order, acknowledging work).

3.07 recognize and minimize legal issues, ethical issues, and potential conflicts of interest in research.

3.08 recognize instances of research misconduct and take steps to address them.

*(Continued)*

**Table 1.** (Continued)

---

**1. Foundational Disciplinary Knowledge** *Understand historical and emerging disciplinary content, concepts, frameworks, and theories and how they relate to other disciplines*.
**Researchers:**

---

3.09 consider the role of social and cultural factors in research.

---

3.10 consider the implications of research to individuals and society.

---

3.11 consider how system structures provide differential access to participation in research.

---

3.12 act to increase access to research for all.

---

**4. Research Communication Skills** *Translate and communicate research ideas and findings in multiple formats to multiple audiences.*
**Researchers:**

---

4.01 construct appropriate ways to present and visualize data.

---

4.02 use disciplinary conventions to communicate research (e.g., ideas, results, implications) orally (e.g., conference presentations, invited talks, research team meetings).

---

4.03 use disciplinary conventions to communicate research (e.g., ideas, results, implications) in writing (e.g., research articles, grant proposals, policy briefs).

---

4.04 translate research findings into policies and practices.

---

4.05 translate research (e.g., ideas, results, implications) and engage with audiences outside of their research discipline (e.g., scholars in other disciplines, non-research, or general audiences).

---

4.06 promote and advocate for research through communications to various audiences (e.g., institutions, disciplines, public stakeholders).

---

**5. Interpersonal Research Skills** *Build relationships and skills to productively interact and collaborate with people from diverse backgrounds and perspectives in the research environment.*
**Researchers:**

---

5.01 understand and conduct themselves in accordance with the cultural and social norms of professionals in the discipline.

---

5.02 express respect for others' differences.

---

5.03 use appropriate and effective interpersonal communication practices with research colleagues.

---

5.04 manage difficult conversations and conflicts with research colleagues.

---

5.05 work effectively with others on collaborative and/or interdisciplinary teams.

---

5.06 consider and include multiple perspectives in decision making.

---

5.07 make meaningful contributions to collaborative research projects.

---

5.08 provide critical and constructive feedback on research to colleagues.

---

5.09 accept, interpret, and modify their research based on constructive criticism and feedback from colleagues.

---

5.10 network with other research professionals.

---

**6. Researcher Self-Beliefs and Attitudes** *Develop personal qualities (i.e., curiosity, confidence, identity, self-regulation, self-assessment and perseverance) that are critical for long-term success in research.*
**Researchers:**

---

6.01 proactively set their research goals and secure the guidance and resources needed to achieve those goals.

---

6.02 persevere when problems or challenges arise in research (e.g., unexpected, ambiguous, or uncertain results, failed projects).

---

6.03 recognize and manage their feelings and behaviors in the research environment.

---

6.04 accurately self-assess their research strengths and weaknesses.

---

6.05 express curiosity in exploring and conducting research.

---

6.06 work at an appropriate level of independence.

---

6.07 manage time to meet individual research project milestones.

---

6.08 develop confidence in their capability to successfully conduct research.

---

6.09 identify themselves as a researcher or expert in their discipline.

---

6.10 engage in practices that support work-life balance (e.g., time management, pursuing interests beyond research).

---

**7. Knowledge and Skills to Pursue a Research or Research-Related Career** *Apply and translate knowledge and skills as a researcher to identify and pursue a research or research-related career.*
**Researchers:**

---

7.01 are aware of career pathways related to their research training.

---

7.02 identify and clarify a long-term strategic vision for their research career.

---

*(Continued)*

**Table 1.** (Continued)

| |
|---|
| **1. Foundational Disciplinary Knowledge** *Understand historical and emerging disciplinary content, concepts, frameworks, and theories and how they relate to other disciplines*. |
| **Researchers:** |
| 7.03 translate and apply their research skills and knowledge across career pathways. |
| 7.04 are prepared to pursue research career pathways. |
| **8. Knowledge and Skills to Administer and Manage Research Projects and Teams** *Develop administrative skills to lead research personnel and projects.* |
| **Researchers:** |
| 8.01 identify and clarify a long-term strategic vision for a program of research. |
| 8.02 identify opportunities and make decisions about the research to be done. |
| 8.03 mentor other developing researchers using best practices in mentoring. |
| 8.04 know how to identify and secure funding (e.g. investments, grants) to support research in their discipline. |
| 8.05 estimate and secure the funds needed to conduct research. |
| 8.06 track research expenditures. |
| 8.07 have the administrative skills to manage research projects and/or heterogenous research teams. |

- **Researcher development frameworks:** evidence-based organization of concepts or ideas around researcher development; research skill assessment.

The scope of the literature search included identifying undergraduate, graduate, and postdoctoral frameworks in the physical, life, or social sciences, arts/humanities or in multiple disciplines. Searches were conducted on Web of Science and Ebsco Academic Search for articles published from 2000 to 2024 using keywords. In the following list of key words, the brackets refer to different iterations of the same search term, the quotes were used to search for specific text, and the asterisk allowed us to search for similar terms: [undergraduate, graduate, postdoctoral] research "development framework" or framework, [undergraduate, graduate, postdoctoral] "research* competencies", "research competency" assessment "research learning" assessment, "research framework" assessment.

In addition, we reviewed professional organization websites for the Council on Undergraduate Research, Council of Graduate Schools, and the National Postdoctoral Association for guidance on key skills that mentees may be expected to learn at each training stage. Finally, frameworks known to us or referenced in articles found in the initial search were incorporated.

## Data evaluation

The articles identified in the literature search were evaluated to determine if they met 3 inclusion criteria: 1) They must define specific areas of knowledge, skill, or psychosocial development; 2) They must address undergraduate, graduate, and/or postdoctoral research training; and 3) They must be published with at least 1 source of validity evidence as defined by the Standards on Educational Testing [18], i.e., evidence-based on test content, response processes, internal structure, or relation to other variables or criterion. Only articles meeting all three criteria were included in the study.

## Analysis & interpretation

The articles that met the inclusion criteria were reviewed, and individual knowledge, skill, and psychosocial elements of researcher development were identified and extracted. Three researchers, a discipline-based educational researcher trained in neurophysiology research methods (JB), an educational psychologist and academic motivation researcher trained in quantitative and qualitative social science research methods (AB), and a director of training and mentoring programs in computational and systems biology and discipline-based educational researcher trained in cellular and

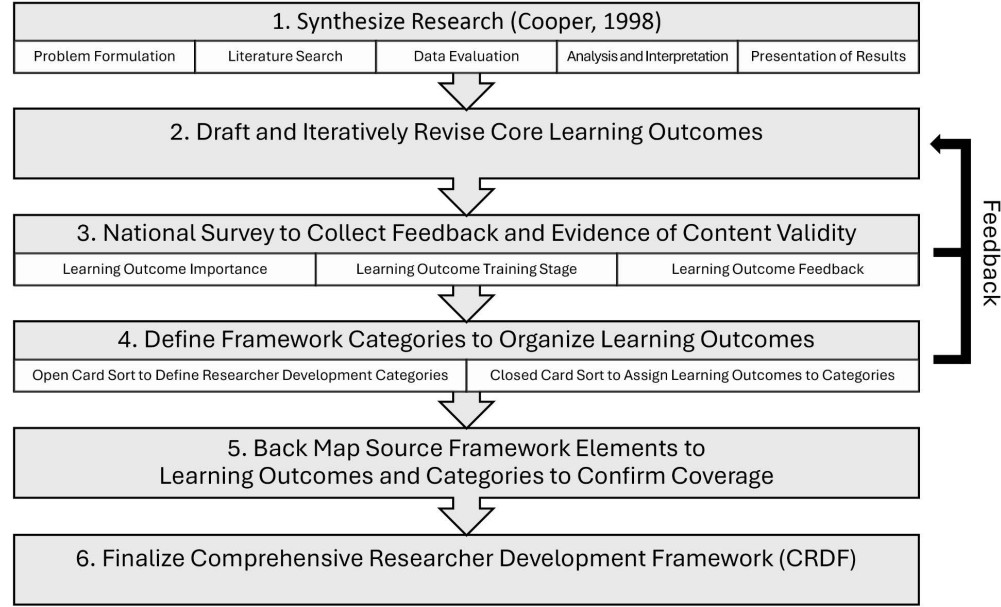

**Fig 1. Comprehensive Researcher Development Framework Development Process.**

developmental neuroscience and molecular genetics (JA), independently analyzed the extracted elements and proposed themes to code the elements through an open coding process [19]. Each researcher organized the elements into similar groups and assigned themes to their groups. The researchers met to compare and discuss the individually proposed group themes and agreed on an initial set of themes and definitions to begin coding. The researchers met to compare, discuss, and revise the themes throughout the coding process. Elements representing general headings or non-research skills (e.g., teaching skills) in the published frameworks were removed from the list of elements.

2. Draft and Iteratively Revise Core Learning Outcomes

One researcher (JB) compared, contrasted, and grouped the elements coded under each of the final themes to draft an initial set of core learning outcomes that represented specific knowledge, skills, and psychosocial attitudes, behaviors, and beliefs. The other two researchers (AB and JA) reviewed the initial drafts, and the team met to discuss their feedback and make revisions. To ensure that all elements extracted from the original source frameworks were represented, JB and AB mapped the original elements to the resulting draft core learning outcomes. Iterative revision of the learning outcomes based on feedback from the research community continued throughout the development process.

3. National Survey to Collect Feedback and Evidence of Content Validity

The draft core learning outcomes were organized into 5 broad categories for use in an online survey to gather feedback from professional researchers (postdoctoral scholars and faculty/staff) across the nation. The broad categories on the survey were: Thinking and Communicating about Research, Researcher Self-Beliefs and Attitudes, Research Career Readiness, Relationships in the Research Environment, and Conducting Research. The S4 Appendix tracks all the learning outcome revisions from the initial to the final versions. While initially designed to capture data on all 79 learning outcomes, 4 learning outcomes related to career pathways were not rated for importance, due to a survey error, and thus results were only available for 75 learning outcomes regarding importance.

The survey was sent to individuals in the researchers' networks, as well as several local, regional, and national groups (see Table in the S5 Appendix for a full listing of groups contacted). Survey respondents were asked to report the level of research mentee(s) with whom they work, their disciplinary area of expertise, their current role, and their institution. To avoid survey fatigue, each respondent was randomly asked one of two questions: 1) How important is achieving each learning outcome to becoming a mature, independent scholar in your discipline? (scale: not relevant to research professionals in my discipline, not important, slightly important, moderately important, extremely important); or 2) At what stage of training in your discipline do research scholars make the greatest gains toward each learning outcome? (scale: not emphasized in training, undergraduate/post baccalaureate, graduate/professional, postdoctoral). Once a respondent answered the first question for each draft learning outcome, they were given the option to answer the other question for each draft learning outcome. After answering one (or both) questions, respondents were invited to give feedback on the draft learning outcomes and to submit any learning outcomes important to their discipline that were missing from the list.

4. Define Framework Categories to Organize Learning Outcomes

Research community members were invited to participate in 2 rounds of card sorting exercises to organize the 78 core learning outcomes into categories for the framework. Card sorting is used to understand how people group and categorize information [20]. Card sorting exercise participants were recruited through email listservs for graduate training program leaders and postdoc scholars at a single research university in the Midwest. The message invited them to sign up and asked them to share the invitation with others in their networks who might be interested in participating.

Open card sorting was conducted in groups that included faculty, research staff, and postdoctoral scholars from multiple disciplines. Participants worked collaboratively to open-sort 78 cards, each with a core learning outcome, into categories defined by the group. First, participants discussed the meaning of each learning outcome, then iteratively organized them into groups, and then named their final groups. Important points of discussion and feedback on the learning outcomes were documented by researchers in real time and used for subsequent revision of the learning outcomes. Data from across groups were combined, and based on analysis of the combined data set, an initial set of categories of researcher development were derived. The categories defined by each group and the cards associated with each group were entered into a spreadsheet that was developed to analyze card sort data [21]. Similar categories offered by groups were merged into standardized categories prior to analysis and the percent agreement among all groups was examined to determine whether the standard categories accurately represented each group of learning outcomes.

Using the categories from the open card sorting exercise, faculty, research staff, postdoctoral scholars, graduate students and undergraduate students from across the country participated in a second, closed card sorting exercise (i.e., all categories were predetermined). Participants either participated in in-person group sessions on the same research university campus as the open card sort exercise or as individuals online using an online card sorting tool [22] at multiple campuses across the country. All were asked to sort the 78 learning outcome cards into the categories generated by the open card sorting exercise. Participants were forced to select one primary category to assign each learning outcome, though many could reasonably be assigned to more than one category, and participants were able to note other categories to which they considered assigning a particular learning outcome. Data from the groups was weighted by the number of participants and combined with the individual card sort data. The combined data was analyzed for patterns to determine under which primary category each learning outcome should nest.

5. Back Map Source Framework Elements to Learning Outcomes and Categories to Confirm Coverage

To ensure that all the elements extracted from the original source frameworks were represented in the final core learning outcomes, the researchers back mapped each extracted element to one or more final learning outcome. The elements were divided into three groups and each researcher mapped one group. Then, a second researcher reviewed the

mapping and either confirmed or flagged it for further discussion. Flagged elements were reviewed and discussed by all researchers to agree on which learning objectives it should be mapped.

## Results

### 1. Synthesize research

The literature search yielded over 13,000 results. Articles were flagged for further review if they addressed undergraduate, graduate or postdoctoral training stages and related to research trainee development. All but 123 articles were excluded from further review because they described empirical research and not researcher learning outcomes or researcher development frameworks.

Of the 123 articles reviewed, 56 met the criteria for inclusion: 34 were identified in either Ebsco or Web of Science, 14 were referenced in other articles, 1 was a professional society framework, and 7 were known to the authors. Thirty-four percent of the articles were published by researchers outside of the United States. The sources, validity evidence, and the discipline(s) and career stage(s) of these 56 frameworks [5,15,23–76] were documented (S1 Appendix). Figs 2 and 3 show summaries of the career stage(s) and disciplines addressed in the articles.

The three researchers reviewed, identified and extracted 1,434 elements from the 56 frameworks. Removal of general headings and non-research skill elements left 1,343 elements to be coded. Each researcher individually reviewed the elements and proposed themes to use in coding. Through discussion, they agreed on an initial shared set of 44 themes to begin coding. Through three rounds of coding, the themes and discrepancies in code assignments were discussed and revisions to the themes and code assignments were made. A final set of 48 themes was used and consensus [19] was reached on the theme codes for all 1,343 elements (45% agreement in the first round, 73% agreement in the second round, and 100% agreement in third round). A full list of the elements and code themes is available in the S3 Appendix.

### 2. Draft and iteratively revise core learning outcomes

One researcher (JB) reviewed the elements coded in each theme and drafted 78 core learning outcomes. These were reviewed by the other two researchers (AB and JA) and the three researchers met to discuss and revise the draft core learning outcomes to generate a revised initial set of 79 core learning outcomes (S4 Appendix).

### 3. National survey to collect feedback and evidence of content validity

To collect feedback on the draft learning outcomes and evidence of content validity (i.e., validity evidence based on the relationship between the content and the construct that it is intended to measure as determined by experts [18]), a national online survey was conducted with researchers asking about the importance of each learning outcome and at what career stage it is emphasized. Overall, 169 individuals responded to the survey. A summary of the characteristics of the survey respondents is presented in Table 2.

### Learning outcome importance

Of our 169 survey respondents, 123 rated how important each learning outcome was to researcher development in their discipline (Fig 4). The overwhelming majority of respondents categorized most of the learning objectives as moderately or extremely important for the development of a researcher in their field. This evidence of content validity from experts supports that the core learning outcomes are relevant to research training across disciplines.

Though there is some variability in level of importance across disciplines, several learning outcomes were consistently rated as extremely important. Table 3 shows the learning outcomes that 80% or more of respondents agreed were extremely important, with perseverance most frequently rated as extremely important.

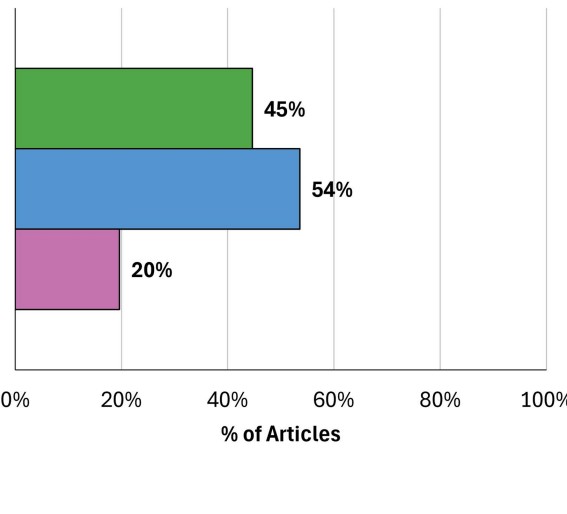

Legend: ■ Undergraduate ■ Graduate ■ Postdoctoral

**Fig 2. Career stages addressed in included framework articles (N = 56).** Note that some articles addressed more than one career stage.

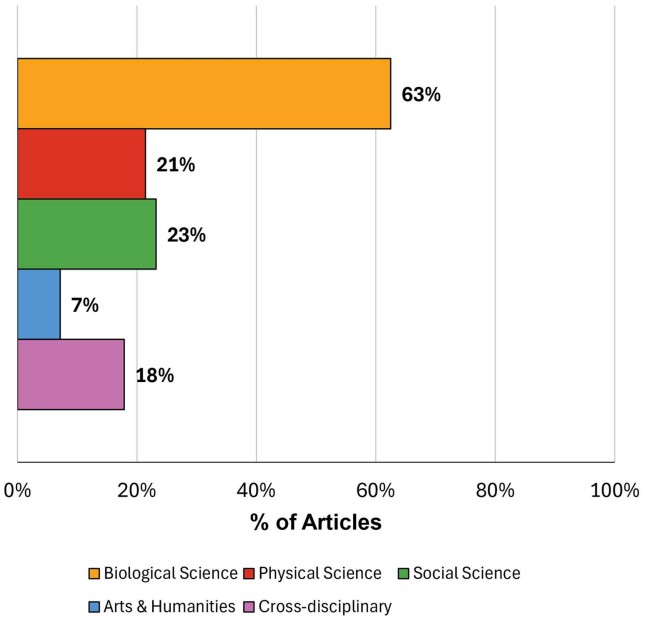

Legend: ■ Biological Science ■ Physical Science ■ Social Science ■ Arts & Humanities ■ Cross-disciplinary

**Fig 3. Research disciplines addressed in included framework articles (N = 56).** Note that some articles addressed more than one discipline.

Overall, two thirds (66.67%) or 50 of the 75 learning outcomes were rated as extremely important by over 50% of the survey respondents. For the remaining 25 (33.33%) there was not a majority rating, nor were there discernable rating patterns based on learning outcome content. We discovered some variation in the level of importance by discipline, with individuals from different disciplines agreeing on the specific level of importance for 29 (38.67%) of the 75 learning outcomes. When we combine the ratings of *moderately important* and *extremely important*, the level of agreement increases to 65 (86.67%).

**Table 2. Characteristics of National Survey Respondents (N = 169).**

| Career Stage | Number of respondents (%) | Discipline | Number of respondents (%) |
|---|---|---|---|
| Research Trainees | 10 (6%) | Biological Sciences | 75 (44%) |
| Faculty/Staff | 102 (60%) | Physical Sciences | 40 (24%) |
| Research Training Program Leaders | 25 (15%) | Social Sciences | 32 (19%) |
| Not reported | 32 (19%) | Arts & Humanities | 9 (5%) |
| | | Cross-Disciplinary | 13 (8%) |

Note: Individuals classified as Cross-Disciplinary included those who ran interdisciplinary training programs as well as training programs that served multiple disciplines.

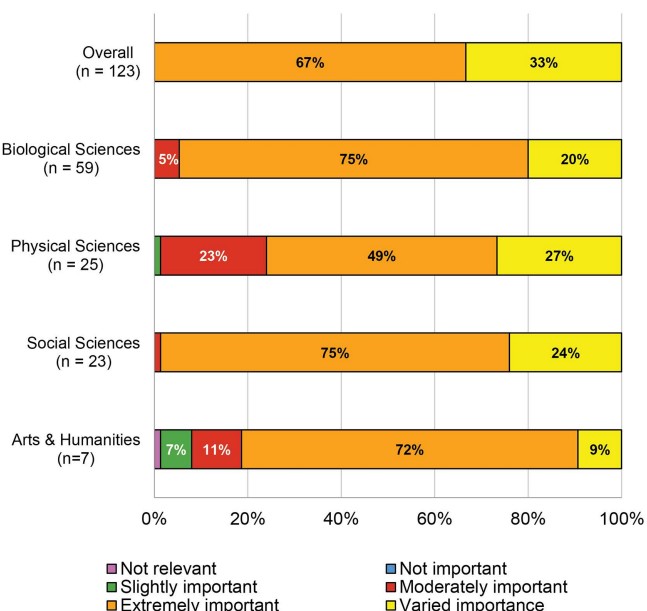

**Fig 4. Percentage of 75 learning outcomes for which 50% or more of respondents selected each level of importance in response to the question "How important is achieving each learning outcome in your discipline?" (N = 123).**

## Learning outcome training stage

We also received input from 112 survey respondents who rated the career stage at which researchers in training make the greatest gains toward each of the 79 learning outcomes in their discipline (Fig 5). Overall, 51 (64.56%) of the learning outcomes were reported as emphasized during graduate education by over 50% of the survey respondents, 7 (8.86%) during undergraduate education, and 6 (7.59%) during postdoctoral training. There was not a majority rating for the remaining 15 (18.99%) learning outcomes, which were reported as addressed during various career stages. When examined within each discipline, trends in the various sciences reflected the overall trend, but the career stage at which learning outcomes were addressed in the Arts & Humanities was primarily shifted to the earlier undergraduate stage. However, given the small number of respondents in the Arts & Humanities it is not possible to draw conclusions about these differences.

Notably, the data show that 7 learning outcomes (3.09, 3.11, 3.12, 4.04, 4.06, 6.10, and 8.06) were rated as not emphasized in training (Fig 5) but were rated as moderately or extremely important (Fig 4) by most respondents. This suggests that research training program directors should consider these important learning outcomes and integrate new learning activities or experiences to address them if their program is not currently addressing them.

**Table 3. Learning Outcomes Rated as Extremely Important by 80% or More of Respondents.**

| Learning Outcome | % Overall Sample (N = 123) |
|---|---|
| persevere when problems or challenges arise in research. | 91% |
| draw conclusions from research results. | 90% |
| interpret or synthesize research results. | 86% |
| can interpret the results of analyses. | 86% |
| use logical and critical thinking in evaluating information and knowledge and in conducting research. | 83% |
| know and follow responsible research practices. | 83% |

Note: The learning outcomes were iteratively revised throughout development based on feedback from the research community. Therefore, some learning outcomes are slightly different than those in the final CRDF (Table 1).

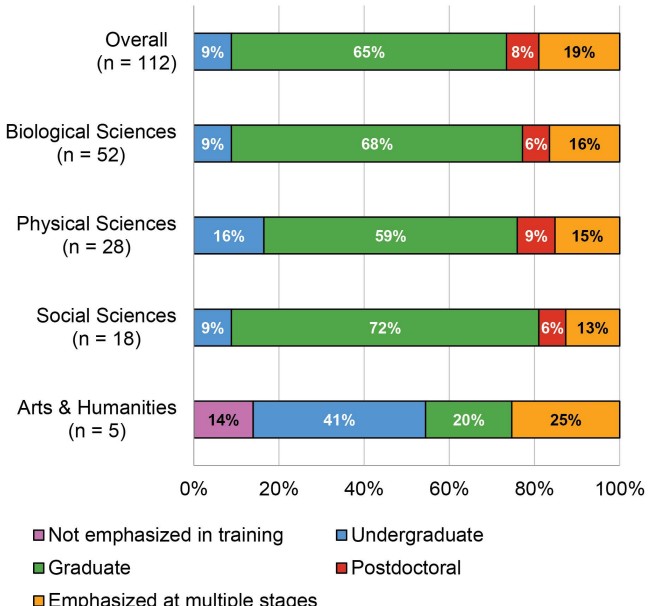

**Fig 5. Percentage of 79 learning outcomes for which 50% or more of respondents selected the career stage at which greatest gains were made. (N = 112).**

In addition to rating the learning outcomes for importance and career stage, survey respondents were asked to provide feedback on the learning outcomes. They were asked to comment on whether the language used was appropriate for their discipline or if any learning outcomes important for their discipline were missing. Based on this feedback and ongoing review by the research team, 21 learning outcomes were revised, 9 were merged, 5 were added, and 1 was deleted. The total number of learning outcomes at the end of this stage of development was 78. See the S4 Appendix for details about the learning outcome revisions made from the initial drafts to the final versions.

4. Define Framework Categories to Organize Learning Outcomes

We solicited input from those in the research community engaged in research training (practitioners and trainees) as well as those who study research training (educational researchers) to organize the learning outcomes into categories (areas

of researcher development) that would be useful to the community. Two separate phases of card sorting activities were implemented. In the first phase, expert research practitioners with deep knowledge of research training (faculty, research staff, and postdoctoral scholars) from across disciplines were recruited to work in groups to open sort the learning outcomes into categories representing areas of researcher development. The preliminary researcher development categories generated across these groups were analyzed and a consensus list developed. In the second phase, professional research practitioners as well as undergraduate and graduate research students used the categories generated from the open sorting exercise to assign the learning outcomes to the categories in a closed sorting exercise. The characteristics of the card sorting exercise participants are summarized in Table 4.

## Open card sort to define researcher development categories

The recruited research professionals worked in nine separate groups of two to four members each to sort the 78 learning outcomes into categories. Each group discussed and named their categories independently. A total of 93 category titles across groups were generated (Table 5) and the category titles were analyzed to identify commonalities. There was overlap, but groups chose to define their categories at different levels of detail. For example, a single category generated

**Table 4. First and Second Round Card Sort Participant Characteristics.**

| | Open Sort – Generate Categories | Closed Sort – Assign Learning Outcomes to Categories | |
| --- | --- | --- | --- |
| | In-person (N = 29) | In-person (N = 16) | Online (N = 46) |
| | N (%) | N (%) | N (%) |
| **Career Stage** | | | |
| Undergraduate Students | 0 (0%) | 5 (31%) | 0 (0%) |
| Graduate Students | 0 (0%) | 6 (38%) | 4 (9%) |
| Postdoctoral Scholars | 13 (45%) | 0 (0%) | 12 (26%) |
| Faculty or Research Staff | 16 (55%) | 5 (31%) | 18 (39%) |
| Not reported | 0 (0%) | 0 (0%) | 12 (26%) |
| **Discipline** | | | |
| Biological Sciences | 19 (66%) | 7 (44%) | 23 (68%) |
| Physical Sciences | 8 (28%) | 8 (50%) | 3 (9%) |
| Social Sciences | 2 (7%) | 7 (44%) | 11 (32%) |
| Arts & Humanities | 1 (3%) | 1 (6%) | 1 (3%) |
| Not reported | | | 12 (26%) |
| **Gender** | | | |
| Woman | 6 (21%) | 7 (44%) | 21 (46%) |
| Man | 7 (24%) | 6 (44%) | 11 (24%) |
| Another gender identity | 0 (0%) | 1 (6%) | 2 (4%) |
| Not reported | 16 (55%) | 1 (6.3%) | 12 (26%) |
| **Race/Ethnicity** | | | |
| White | 11 (38%) | 11 (69%) | 27 (59%) |
| Non-White | 2 (7%) | 5 (31%) | 7 (15%) |
| Not reported | 16 (55%) | 0 (0%) | 12 (26%) |

Note: Some participants reported more than one discipline, Gender, or Race/Ethnicity. Gender and Race/Ethnicity information was collected as part of a follow up survey for Open Card sort participants, so higher numbers of not reported information are present in those cells. Non-White included individuals who identified as American Indian or Alaskan Native, Asian, Hispanic or Latinx, or with Two or More Races. Another gender identity includes individuals who reported their gender as genderqueer, non-binary, or transgender. The numbers in these categories are reported together to protect participants' confidentiality.

**Table 5. Evolution of Area of Researcher Development Categories.**

| Open Card Sort Categories Generated by Different Groups | | | Integrated Preliminary Categories | | Final Categories After Closed Sort |
|---|---|---|---|---|---|
| • Research Content Knowledge<br>• Foundational knowledge<br>• Expertise – Early Stage | • Discipline/ Disciplinary Knowledge<br>• Knowledge/ breadth | → | **Foundational Disciplinary Knowledge** | | |
| • Research Interpretation<br>• Synthesis<br>• Interpretation<br>• Expertise – Late Stage<br>• Experimental Design & Analysis<br>• Research Skills Related to Overall Discipline/ Research Vision | • Critical thinking<br>• Research independence/ novel research questions<br>• Basic Research Skills<br>• Research process – conducting research & synthesizing new knowledge | → | *Research Thinking and Reasoning Skills* | → | **Practical and Cognitive Research Skills** |
| • Research Tech Skills<br>• Planning/ hypothesis<br>• Technical Competency<br>• Expertise – Mid Stage<br>• Foundational Research Skills | • Technical Research Skills<br>• How to Design and Execute an Experiment or Project<br>• Technical<br>• Analysis/technical skills | → | *Practical Research Skills* | | |
| • Ethics/ Ethical Practices/Research ethics<br>• Ethical/ Professional Behavior<br>• Guidelines/ Professional Ethics<br>• Safety | • Responsible Conduct of research/ Research Integrity<br>• Broader Impacts | → | **Ethical and Responsible Research Practices** | | |
| • Translational/ Public Science<br>• Professional Communication<br>• Communication & Sharing to Society<br>• Public/ Broad Communication | • Research Communication<br>• Scientific Communication<br>• Scientific Communication to non-scientific community | → | **Research Communication Skills** | | |
| • Social aspects/ lab culture<br>• Communication w/ others<br>• Access/Inclusion<br>• Communication/ Feedback<br>• Teams/ Group Networking<br>• Mentoring<br>• People (Interpersonal)<br>• Collaborative Research<br>• Promoting Equality in Research<br>• Professional Collaboration<br>• Mentorship | • Professional Skills<br>• Inclusion<br>• Collaboration<br>• Management<br>• Communication and collaboration<br>• Mentorship soft skills management<br>• Advocacy<br>• Collaborative Teams<br>• Leadership<br>• Interpersonal & research Management | → | *Interpersonal Skills as a Researcher* | → | **Interpersonal Research Skills** |
| • Self-efficacy<br>• Affective<br>• Agency, Grit, Ownership, Resilience<br>• Grit/Self-Regulation | • Attitude/Personal Development<br>• Self-Management<br>• Self-efficacy & Research identity | → | *Personal Attributes as a Researcher* | → | **Researcher Self-Beliefs and Attitudes** |
| • Career exploration<br>• Career<br>• Research/Career Independence<br>• Career Pathway | • Personal/Career Knowledge Development<br>• Career Development | → | **Knowledge and Skills to Pursue a Research or Research-Related Career** | | |
| • Administration<br>• Money/ Resources – Research Admin<br>• Finance Resources<br>• Research Management & Implementation<br>• Management/Administrative | • Funding<br>• Money<br>• Lab Management<br>• Funding | → | *Knowledge and Skills to Manage Research Projects and Teams* | → | **Knowledge and Skills to Administer and Manage Research Projects and Teams** |

Closed Sort to Assign Learning Outcomes to Categories.

by one group could align with multiple categories generated by another group. Comparisons of the learning outcomes each group assigned to their categories were used to understand how categories across different groups were related and to identify consensus categories across groups. Based on this data analysis, the research team defined nine integrated preliminary categories (Table 5). Results from the individual groups are available in S6 Appendix.

Faculty, research staff, postdoctoral scholars, graduate students, and undergraduate students participated in the closed card sorting exercise to provide feedback and assign the 78 learning outcomes to the 9 integrated preliminary categories generated by the open card sorting exercise (Table 5). Some participants engaged as part of in-person groups while others engaged individually with an online card sorting tool. Nine in-person groups with two to four participants each were held, and 46 individuals from across the country participated online. Data from the groups was weighted by the number of individuals in the group and combined with online data from individuals. For example, if four individuals participated in a focus group, the results of their card sort were weighted four times compared to card sort data from individual participants who completed the online card sort activity.

A map of the closed card sorting results is presented in Table 6. Overall, closed card sorting participants reported that it was sometimes challenging to assign a learning outcome to just one area. Consensus across groups for each learning outcome shown in Table 6 was considered strong if there was 75–100% agreement (dark blue), moderate if there was 50–75% agreement (light blue), and weak if there was less than 50% agreement (no shading). Assignment of learning outcomes between the "Research Thinking and Reasoning Skills" and "Practical Research Skills" categories showed significant overlap. Therefore, these two categories were combined into one, "Practical and Cognitive Research Skills," and the data from the two original categories were combined for analysis.

After the combined "Practical and Cognitive Research Skills" category was created, analysis of the raw data showed strong agreement for 8 (36%) of the 24 learning outcomes and moderate agreement for 13 (59%). These were assigned to the given category, and no further modifications were made to them. Learning outcomes across all categories that showed weak agreement were reviewed by the research team. Those that were misinterpreted based on observations during in-person card sorting sessions or comments shared by online participants were revised to clarify their meaning. Those that generated equally valid, but different interpretations by different groups or individuals were divided into two separate learning outcomes. Development and refinement of the core learning outcomes are detailed in the S4 Appendix.

In addition to combining the two areas of researcher development, the names of three other areas of researcher development were modified based on the closed card sort participant feedback. "Interpersonal Skills as a Researcher" was modified to "Interpersonal Research Skills" to clarify that this category includes learning outcomes focused on a researcher's ability to interact with other researchers. "Personal Attributes as a Researcher" was modified to "Researcher Self-Beliefs and Attitudes" based on feedback received that the word "attribute" suggested that these were unchangeable traits, rather than psychosocial skills that individuals can develop. Finally, "Knowledge and Skills to Manage Research Projects and Teams" was modified to "Knowledge and Skills to Administer and Manage Research Projects and Teams" to clarify that this category included learning outcomes needed for administration of research. The last column in Table 5 shows the final categories. The final CRDF with the categories and their nested core learning outcomes is in Table 1.

5. Back Map Source Framework Elements to Learning Outcomes and Categories to Confirm Coverage

Researchers back mapped the elements derived from the original source frameworks to the final CRDF core learning outcomes and areas of researcher development (categories) to confirm that all were covered in the final CRDF. Table 7 reports the percentage of CRDF core learning outcomes in each area of researcher development addressed in the original source frameworks. These percentages were calculated from mapping each source element to one or more core learning outcomes. The detailed map of each element to a core learning objective(s) is in the S7 Appendix.

**Table 6. Phase 2 Card Sorting Results.**

| Areas of Researcher Development and Learning Outcomes | % Agreement |
|---|---|
| **Foundational Disciplinary Knowledge** | |
| know the fundamental content in their discipline (e.g., frameworks, theories). | 83.9% |
| can relate content knowledge from other disciplines to content knowledge in their discipline[1]. | 53.2% |
| ground hypotheses and research questions in established disciplinary knowledge, theories, or frameworks. | 46.8% |
| know the history of knowledge generation in their discipline. | 85.5% |
| know the processes by which new knowledge is generated and evaluated in their discipline. | 72.6% |
| **Practical and Cognitive Research Skills** | |
| know assumptions and limitations in study designs (e.g., reporting uncertainty/error). | 66.1% |
| select appropriate methods to investigate research questions in the discipline. | 71.0% |
| draw conclusions from research results. | 88.7% |
| use disciplinary theories, frameworks and models in analyzing the results of research studies. | 62.9% |
| use logical and critical thinking in evaluating information in research (e.g., designing, conducting, defending, and reviewing research) | 71.0% |
| consider alternative approaches and interpretations of research. | 75.8% |
| set research goals. | 54.8% |
| refine existing and/or contribute new disciplinary theories, frameworks, and models. | 50.0% |
| connect diverse research ideas and approaches in novel ways. | 69.4% |
| interpret or synthesize research results. | 79.0% |
| can interpret the results of analyses of data (e.g., coding, mathematical and statistical calculations) | 83.9% |
| can provide a logical rationale for their study designs. | 67.7% |
| recognize the inferences and implications of research findings on and beyond the discipline[1]. | 38.7% |
| identify gaps in existing knowledge or research results to investigate. | 56.5% |
| use disciplinary theories, frameworks and models in designing research studies. | 53.2% |
| formulate hypotheses and research questions that can be systematically tested or investigated. | 74.2% |
| have up-to-date technical skills to conduct research in the discipline. | 79.0% |
| use troubleshooting skills to address theoretical or technical problems in research. | 88.7% |
| can use tools and databases to search the disciplinary literature. | 61.3% |
| select the appropriate analytic and statistical methods used in the discipline. | 83.9% |
| use literature search strategies that identify relevant prior research. | 53.2% |
| develop new data collection or analytical methods when needed to address novel research questions. | 82.3% |
| **Ethical and Responsible Research Practices** | |
| recognize and minimize potential conflicts of interest in research | 87.1% |
| know and follow disciplinary norms and policies regarding credit for contributions to research (e.g., citing previous research, authorship order, acknowledging work). | 72.6% |
| understand how system structures provide differential access to participation in research[2]. | 30.6% |
| follow standard protocols to document and securely store research data[1]. | 46.8% |
| consider the implications of research to individuals and society. | 54.8% |
| recognize instances of research misconduct and take steps to address them | 82.3% |
| consider the role of social and cultural factors in research. | 61.3% |
| know and follow guidelines for ethical treatment of research subjects (e.g., individuals, communities, animals, etc.) | 83.9% |
| know and follow guidelines for research rigor and reproducibility in your discipline | 62.9% |
| know and follow disciplinary data ownership/stewardship practices | 69.4% |
| follow research safety regulations. | 59.7% |
| act to increase access to research for all. | 27.4% |
| **Research Communication Skills** | |
| use disciplinary conventions to communicate research effectively (e.g., ideas, results, implications) orally (e.g., conference presentations, invited talks, research team meetings). | 85.5% |

*(Continued)*

**Table 6.** (Continued)

| Areas of Researcher Development and Learning Outcomes | % Agreement |
|---|---|
| construct appropriate ways to present and visualize data. | 61.3% |
| promote and advocate for research within the institution, the discipline, and through interactions with public stakeholders[2]. | 46.8% |
| are able to translate research findings into policies, practices, and daily life. | 54.8% |
| can translate research (e.g., ideas, results, implications) and engage with audiences outside of their research discipline (e.g., to scholars in other disciplines, non-research, or general audiences). | 77.4% |
| use disciplinary conventions to communicate research effectively (e.g., ideas, results, implications) in writing (e.g., research articles, grant proposals, policy briefs). | 67.7% |
| Interpersonal Skills as a Researcher | |
| accept, interpret, and modify their research based on constructive criticism and feedback from colleagues. | 35.5% |
| make meaningful contributions to collaborative research projects. | 58.1% |
| consider and include multiple perspectives in decision making. | 54.8% |
| use appropriate and effective interpersonal communication practices with research colleagues. | 77.4% |
| understand and conduct themselves in accordance with the cultural and social norms of professionals in the discipline. | 45.2% |
| work effectively with others on collaborative and/or interdisciplinary teams. | 80.6% |
| express respect for others' differences. | 69.4% |
| are able to network with other research professionals. | 64.5% |
| provide critical and constructive feedback on research to colleagues. | 62.9% |
| are able to manage difficult conversations and conflicts with research colleagues. | 62.9% |
| Personal Attributes as a Researcher | |
| develop confidence in their capability to successfully conduct research. | 77.4% |
| engage in practices that support work-life balance (e.g., time management, pursuing interests beyond research) | 69.4% |
| persevere when problems or challenges arise in research (e.g., unexpected, ambiguous or uncertain results, failed projects) | 80.6% |
| are able to accurately self-assess their strengths and weaknesses. | 83.9% |
| develop attitudes about research that support success in research. | 64.5% |
| are able to recognize and manage their feelings and behaviors in the research environment. | 75.8% |
| identify themselves as a researcher or expert in their discipline. | 79% |
| work at an appropriate level of independence. | 61.3% |
| express curiosity in exploring and conducting research. | 67.7% |
| self-advocate and take responsibility when working with mentors to set research goals and secure the guidance and resources needed to achieve those goals. | 66.1% |
| are able to manage time and meet research project milestones in a timely manner[2]. | 41.9% |
| Knowledge and Skills to Pursue a Research or Research-Related Career | |
| are aware of career pathways related to their research training. | 88.7% |
| are prepared to pursue research career pathways. | 83.9% |
| can translate and apply research skills and knowledge across career pathways. | 72.6% |
| Knowledge and Skills to Manage Research Projects and Teams | |
| identify and clarify a long-term strategic vision for research[1]. | 35.5% |
| mentor other developing researchers using best practices in mentoring. | 51.6% |
| can estimate the funds needed to conduct research. | 67.7% |
| are able to manage heterogenous research teams. | 82.3% |
| know how research is funded in the discipline[2]. | 29% |
| identify opportunities and make decisions about the research to be done[2]. | 27.4% |
| can track research expenditures. | 74.2% |
| have the administrative skills to manage research projects, personnel, and support staff. | 77.4% |
| can secure funding to conduct research[3]. | 45.2% |

Note: [1] learning outcome was split into multiple learning outcomes; [2] learning outcome was revised; [3] learning outcome was removed.

## Discussion

To our knowledge, the CRDF presented here is the first published framework that synthesizes common outcomes across multiple disciplines and addresses undergraduate through postdoctoral training career stages. The CRDF will play an important role in standardizing programmatic, assessment and evaluation efforts, as well as demystify the researcher development process for mentees and mentors. Though feedback and evidence of validity were gathered from the research community in the United States, 34% of the source frameworks were published by scholars outside the United States suggesting the CRDF will be applicable globally. Below we detail needs that this new framework will address and how it will benefit the research community at large.

### The CRDF will identify focus areas and potential gaps in existing frameworks

Currently available frameworks tend to address only a subset of the areas of researcher development and be discipline and/or career stage specific. Focus areas and gaps in the 56 frameworks used to develop the new framework are revealed in Table 7, where the elements of these source frameworks are mapped to the eight areas of researcher development in the CRDF and in the S7 Appendix where they have been mapped in greater detail to the individual 79 learning outcomes. For example, the map in Table 7 reveals that several frameworks include a focus on Research Communication Skills, while many fewer frameworks address developing Knowledge and Skills to Pursue a Research or Research-Related Career. Using the CRDF, research training program directors can compare any framework they're using to design and/or evaluate their training programs to the CRDF to identify focus areas and gaps they may need to address.

### The CRDF will align the training and performance expectations of multiple stakeholders engaged in research training

The learning outcomes in the CRDF are meant to represent a relative consensus across disciplines to promote a shared understanding of the knowledge, skills, and psychosocial attitudes, behaviors, and beliefs that should be developed through research training among the multiple stakeholders (e.g., mentees, mentors, training programs, disciplinary communities, funders) involved in developing researchers and the research workforce. The core learning outcomes provide the means to track mentee development across training career stages and disciplines, as well as the structure needed to coordinate training across institutions. Training programs that use the CRDF to design their programs and assess development of their mentees will be better positioned to collaborate and provide support for mentees in transition from one program to the next and therefore contribute to national efforts to develop the research workforce. Those in STEM will also be better positioned to address the recommendations outlined in two recent National Academies of Science, Engineering and Medicine (NASEM) reports on undergraduate [3] and graduate research [4] training, which were referenced in developing the CRDF.

### The CRDF will level the playing field for mentees by making the expectations in research training transparent

The lack of structure in research learning experiences, especially apprentice-style research learning experiences, has created an irregular and hidden curriculum that disadvantages students with limited research backgrounds (e.g., first-generation college students). Novice research mentees often lack the knowledge and social capital they need to successfully navigate the research environment and are consequently more likely to encounter and struggle to meet expectations and navigate unanticipated challenges throughout their training journey [77–80]. The core learning outcomes in the CRDF clarify what mentees should be learning during formal research training and can therefore empower them to meet expectations and successfully navigate the research training environment. The core outcomes also build mentees' agency to take responsibility in designing their training journey by allowing them to self-assess their progress and to self-advocate for learning experiences that will support achievement of the learning outcomes. The CRDF can be shared with research

**Table 7. Original Framework Elements Mapped to Areas of Researcher Development.**

| Frame-work Number | Framework Article | Foun-dational Disci-plinary Knowl-edge | Practi-cal and Cognitive Research Skills | Ethical and Respon-sible Research Practices | Research Commu-nication Skills | Interper-sonal Research Skills | Researcher Self-Beliefs and Attitudes | Knowledge/Skills to Pursue a Research or Research-Related Career | Knowledge/Skills to Administer and Manage Research Projects/Teams |
|---|---|---|---|---|---|---|---|---|---|
| 2 | Competency-based assessment for the training of PhD students and early-career scientists. [23] | 50.0% | 29.2% | 33.3% | 50.0% | 30.0% | 20.0% | 50.0% | 42.9% |
| 3 | Researcher Skill Development Framework (US English Edition) [24] | 0.0% | 25.0% | 8.3% | 33.3% | 20.0% | 0.0% | 0.0% | 14.3% |
| 4 | National Postdoctoral Association Core Competencies [5] | 33.3% | 33.3% | 58.3% | 66.7% | 60.0% | 20.0% | 25.0% | 71.4% |
| 5 | The Basic Competencies of Biological Experimentation: Concept-Skill Statements. [25] | 50.0% | 70.8% | 33.3% | 66.7% | 0.0% | 0.0% | 0.0% | 14.3% |
| 6 | Development and National Validation of a Tool for Interpret-ing the Vision and Change Core Competencies [26] | 50.0% | 79.2% | 91.7% | 100.0% | 50.0% | 20.0% | 0.0% | 0.0% |
| 7 | Qualitative Investigation to Identify the Knowledge and Skills That U.S.-Trained Doctoral Chemists Require in Typical Chemistry Positions [27] | 16.7% | 29.2% | 33.3% | 50.0% | 30.0% | 20.0% | 0.0% | 28.6% |
| 8 | Assessment in Undergraduate Research: The EvaluateUR Method [28] | 66.7% | 33.3% | 25.0% | 33.3% | 60.0% | 50.0% | 75.0% | 0.0% |
| 9 | Entering Research Learning Assessment (ERLA): Validity Evi-dence for an Instrument to Mea-sure Undergraduate and Graduate Research Trainee Development. [15] | 16.7% | 20.8% | 16.7% | 100.0% | 30.0% | 30.0% | 75.0% | 28.6% |
| 10 | Towards a framework for research career development: An evalua-tion of the UK's Vitae Researcher Development Framework. [29] | 16.7% | 8.3% | 0.0% | 33.3% | 20.0% | 10.0% | 50.0% | 42.9% |
| 11 | A competency framework for Ph.D. programs in health information management. [30] | 0.0% | 25.0% | 0.0% | 33.3% | 0.0% | 0.0% | 0.0% | 0.0% |
| 12 | Research competencies for under-graduate rehabilitation students: a scoping review [31] | 50.0% | 45.8% | 25.0% | 50.0% | 10.0% | 30.0% | 0.0% | 14.3% |
| 13 | Competency-based postdoc-toral research training for clinical psychologists: An example and implications. [32] | 50.0% | 12.5% | 16.7% | 50.0% | 10.0% | 10.0% | 0.0% | 42.9% |
| 14 | Commonly known, commonly not known, totally unknown: a framework for students becoming researchers [33] | 0.0% | 20.8% | 0.0% | 33.3% | 0.0% | 0.0% | 0.0% | 0.0% |

*(Continued)*

**Table 7.** (Continued)

| Frame-work Number | Framework Article | Foundational Disciplinary Knowledge | Practical and Cognitive Research Skills | Ethical and Responsible Research Practices | Research Communication Skills | Interpersonal Research Skills | Researcher Self-Beliefs and Attitudes | Knowledge/Skills to Pursue a Research or Research-Related Career | Knowledge/Skills to Administer and Manage Research Projects/Teams |
|---|---|---|---|---|---|---|---|---|---|
| 15 | Development of the Scientific Research Competency Scale for nurses [34] | 33.3% | 37.5% | 50.0% | 33.3% | 20.0% | 50.0% | 25.0% | 57.1% |
| 16 | Social-Scientific Research Competency. [35] | 0.0% | 29.2% | 0.0% | 0.0% | 0.0% | 20.0% | 25.0% | 28.6% |
| 17 | Evaluating the development of chemistry undergraduate researchers' scientific thinking skills using performance-data: first findings from the performance assessment of undergraduate research (PURE) instrument. [36] | 16.7% | 37.5% | 0.0% | 16.7% | 0.0% | 0.0% | 0.0% | 0.0% |
| 18 | Assessment of Undergraduate Research Learning Outcomes: Poster Presentations as Artifacts. [37] | 0.0% | 12.5% | 16.7% | 33.3% | 10.0% | 10.0% | 0.0% | 0.0% |
| 19 | Objectivity of the subjective quality: Convergence on competencies expected of doctoral graduates [38] | 0.0% | 0.0% | 0.0% | 16.7% | 0.0% | 10.0% | 0.0% | 0.0% |
| 20 | Experiences of using the researching professional development framework [39] | 0.0% | 4.2% | 0.0% | 50.0% | 10.0% | 20.0% | 0.0% | 14.3% |
| 21 | Developing a Competency Framework for Population Health Graduate Students Through Student and Faculty Collaboration. [40] | 16.7% | 0.0% | 0.0% | 83.3% | 0.0% | 0.0% | 0.0% | 14.3% |
| 22 | Professional learning and development framework for postdoctoral scholars [41] | 0.0% | 4.2% | 8.3% | 100.0% | 20.0% | 20.0% | 50.0% | 28.6% |
| 23 | Development and psychometric testing of the Research Competency Scale for Nursing Students: An instrument design study [42] | 16.7% | 29.2% | 8.3% | 33.3% | 0.0% | 0.0% | 0.0% | 0.0% |
| 24 | Bioinformatics core competencies for undergraduate life sciences education. [43] | 33.3% | 20.8% | 25.0% | 0.0% | 0.0% | 0.0% | 0.0% | 0.0% |
| 25 | A systematic review of doctoral graduate attributes: Domains and definitions [44] | 33.3% | 33.3% | 25.0% | 100.0% | 20.0% | 70.0% | 0.0% | 14.3% |
| 26 | A structured professional development curriculum for postdoctoral fellows leads to recognized knowledge growth [45] | 0.0% | 0.0% | 8.3% | 0.0% | 10.0% | 20.0% | 0.0% | 28.6% |
| 27 | Guidelines for competency development and measurement in rehabilitation psychology postdoctoral training [46] | 16.7% | 25.0% | 16.7% | 33.3% | 0.0% | 0.0% | 0.0% | 0.0% |

*(Continued)*

**Table 7.** (Continued)

| Frame-work Number | Framework Article | Foun-dational Disci-plinary Knowl-edge | Practi-cal and Cognitive Research Skills | Ethical and Respon-sible Research Practices | Research Commu-nication Skills | Interper-sonal Research Skills | Researcher Self-Beliefs and Attitudes | Knowledge/ Skills to Pursue a Research or Research-Related Career | Knowledge/ Skills to Administer and Manage Research Projects/ Teams |
|---|---|---|---|---|---|---|---|---|---|
| 28 | Are You Doing It Backward? Improving Information Literacy Instruction Using the AALL Prin-ciples and Standards for Legal Research Competency, Taxono-mies, and Backward Design. [47] | 66.7% | 58.3% | 33.3% | 50.0% | 10.0% | 10.0% | 0.0% | 28.6% |
| 29 | Evaluating research-oriented teaching: a new instrument to assess university students' research competences. [48] | 33.3% | 29.2% | 8.3% | 33.3% | 0.0% | 0.0% | 0.0% | 0.0% |
| 30 | Research skills for university students' thesis in E-learning: Scale development and validation in Peru [49] | 16.7% | 20.8% | 0.0% | 16.7% | 0.0% | 0.0% | 0.0% | 0.0% |
| 31 | Evaluating Undergraduate Research Experiences—Develop-ment of a Self-Report Tool. [50] | 66.7% | 41.7% | 0.0% | 50.0% | 20.0% | 10.0% | 0.0% | 0.0% |
| 32 | Evaluating a Summer Undergradu-ate Research Program: Measuring Student Outcomes and Program Impact. [51] | 50.0% | 41.7% | 25.0% | 33.3% | 40.0% | 50.0% | 0.0% | 0.0% |
| 33 | Threshold concepts in research education and evidence of thresh-old crossing [52] | 33.3% | 20.8% | 0.0% | 0.0% | 0.0% | 0.0% | 0.0% | 0.0% |
| 34 | Faculty Mentors', Graduate Stu-dents', and Performance-Based Assessments of Students' Research Skill Development [54] | 16.7% | 25.0% | 0.0% | 0.0% | 0.0% | 0.0% | 0.0% | 14.3% |
| 35 | Postdocs' lab engagement predicts trajectories of PhD students' skill development. [54] | 0.0% | 29.2% | 8.3% | 33.3% | 0.0% | 0.0% | 0.0% | 0.0% |
| 36 | Development of the Research Competencies Scale [55] | 50.0% | 50.0% | 50.0% | 33.3% | 0.0% | 0.0% | 0.0% | 0.0% |
| 37 | Competency-Based Postdoctoral Education [56] | 16.7% | 33.3% | 8.3% | 66.7% | 10.0% | 0.0% | 0.0% | 14.3% |
| 38 | An Exploratory Investigation of the Research Self-Efficacy, Inter-est in Research, and Research Knowledge of Ph.D. in Education Students [57] | 16.7% | 12.5% | 8.3% | 33.3% | 0.0% | 0.0% | 0.0% | 0.0% |
| 39 | Inquiry Experiences to the NACE Career Readiness Competencies [58] | 0.0% | 29.2% | 25.0% | 66.7% | 50.0% | 40.0% | 25.0% | 14.3% |
| 40 | Development and implementation of a competency-based module for teaching research methodology to medical undergraduates [59] | 0.0% | 33.3% | 8.3% | 16.7% | 20.0% | 0.0% | 0.0% | 0.0% |

*(Continued)*

| Frame-work Number | Framework Article | Foun-dational Disci-plinary Knowl-edge | Practi-cal and Cognitive Research Skills | Ethical and Respon-sible Research Practices | Research Commu-nication Skills | Interper-sonal Research Skills | Researcher Self-Beliefs and Attitudes | Knowledge/ Skills to Pursue a Research or Research-Related Career | Knowledge/ Skills to Administer and Manage Research Projects/ Teams |
|---|---|---|---|---|---|---|---|---|---|
| 41 | Development of a structured undergraduate research experi-ence: Framework and implications. [60] | 0.0% | 8.3% | 0.0% | 33.3% | 20.0% | 10.0% | 0.0% | 28.6% |
| 42 | Criteria for academic bachelor's and master's curricula. [61] | 100.0% | 62.5% | 16.7% | 50.0% | 30.0% | 70.0% | 25.0% | 0.0% |
| 43 | Doctoral conceptual thresholds in cellular and molecular biology. [62] | 0.0% | 8.3% | 0.0% | 0.0% | 0.0% | 0.0% | 0.0% | 0.0% |
| 44 | Toward a conceptual framework for measuring the effectiveness of course-based undergraduate research experiences in under-graduate biology. [63] | 16.7% | 4.2% | 0.0% | 100.0% | 10.0% | 0.0% | 0.0% | 0.0% |
| 45 | Building sustainability research competencies through scaffolded pathways for undergraduate research experience. [64] | 16.7% | 4.2% | 0.0% | 0.0% | 20.0% | 10.0% | 0.0% | 14.3% |
| 46 | Developing Research Compe-tence to Support Evidence-Based Practice. [65] | 0.0% | 33.3% | 0.0% | 66.7% | 0.0% | 0.0% | 0.0% | 14.3% |
| 47 | Evaluator Competencies: What's Taught Versus What's Sought. [66] | 16.7% | 25.0% | 0.0% | 66.7% | 0.0% | 0.0% | 0.0% | 14.3% |
| 48 | Evaluating Mastery of Biostatistics for Medical Researchers: Need for a New Assessment Tool [67] | 0.0% | 29.2% | 8.3% | 33.3% | 0.0% | 0.0% | 0.0% | 0.0% |
| 49 | Application of the competency model to clinical health psychology [68] | 33.3% | 12.5% | 8.3% | 33.3% | 20.0% | 0.0% | 0.0% | 0.0% |
| 50 | Competency-Based Veterinary Education and Assessment of the Professional Competencies [69] | 16.7% | 4.2% | 0.0% | 0.0% | 0.0% | 20.0% | 0.0% | 0.0% |
| 51 | Climbing the stairway to compe-tency: Trainee perspectives on competency development. [70] | 16.7% | 29.2% | 0.0% | 33.3% | 0.0% | 0.0% | 0.0% | 0.0% |
| 52 | MIA Board white paper: definition of biomedical informatics and specification of core competen-cies for graduate education in the discipline [71] | 66.7% | 41.7% | 8.3% | 16.7% | 10.0% | 0.0% | 0.0% | 42.9% |
| 53 | Building Interdisciplinary Research Models: A Didactic Course to Prepare Interdisciplinary Scholars and Faculty [72] | 16.7% | 12.5% | 0.0% | 66.7% | 60.0% | 0.0% | 0.0% | 0.0% |
| 54 | Applying the Cube Model to Pedi-atric Psychology: Development of Research Competency Skills at the Doctoral Level [73] | 66.7% | 20.8% | 16.7% | 33.3% | 30.0% | 0.0% | 0.0% | 0.0% |

*(Continued)*

**Table 7.** (Continued)

| Frame-work Number | Framework Article | Foundational Disciplinary Knowledge | Practical and Cognitive Research Skills | Ethical and Responsible Research Practices | Research Communication Skills | Interpersonal Research Skills | Researcher Self-Beliefs and Attitudes | Knowledge/ Skills to Pursue a Research or Research-Related Career | Knowledge/ Skills to Administer and Manage Research Projects/ Teams |
|---|---|---|---|---|---|---|---|---|---|
| 55 | Information Literacy Competency Standards for Higher Education. [74] | 16.7% | 50.0% | 41.7% | 50.0% | 10.0% | 0.0% | 0.0% | 14.3% |
| 56 | Developing a Scoring Rubric for Resident Research Presentations: A Pilot Study [75] | 0.0% | 16.7% | 0.0% | 16.7% | 10.0% | 0.0% | 0.0% | 0.0% |
| 57 | Core Competencies for Research Training in the Clinical Pharmaceutical Sciences [76]] | 33.3% | 25.0% | 16.7% | 16.7% | 0.0% | 0.0% | 0.0% | 14.3% |

mentees by including it in a student handbook or by using the Researcher Development Plan tool provided in the S2a and S2b Appendix, which can be implemented in conjunction with Individual Development Plans and Mentor-Mentee Compacts.

## The CRDF will support the development of common metrics to measure and understand how researchers develop across training programs

Core learning outcomes synthesized in the CRDF provide the basis for developing measurement tools such as learning assessments, rubrics, and interview/focus group protocols that can be used in program evaluation and basic research on research training and researcher development. Evaluators can use the data generated by common metrics to provide targeted feedback to research training program directors and mentors about the efficacy of the research learning experiences they are providing to guide the continuous improvement of specific learning experiences for mentees and research training programs [81]. Researchers can use the tools in their investigations of researcher development and research learning experiences across multiple training sites, thus potentially discovering causal relationships and generalizable knowledge about researcher development that can be applied broadly. Common metrics used across multiple sites allow investigation of the mechanisms by which research training environments, specific learning experiences, and individual mentee characteristics contribute to or inhibit researcher development. Even when common metrics or frameworks are not used across programs, the CRDF may allow researchers and practitioners to align outcomes from different studies to investigate the impact of research training programs [3,82–84].

## Limitations

We were able to gather only limited input on the CRDF from the arts and humanities research community. Therefore, it is difficult to draw conclusions about the use of the framework in this community or about the disciplinary differences we observed between this community and the sciences. Nonetheless, since arts and humanities frameworks were used to construct the CRDF and we are confident in the feedback we did receive, we believe the CRDF can be used in these disciplines. Further testing in this community is needed to verify or highlight areas for revision in the framework for use in arts and humanities.

The majority, though not all the content validity data reported here was from one large research university. Likewise, there was a lack of racial/ethnic diversity in the national survey respondent pool and among the card sorting activity participants for whom we have this information. Therefore, as the framework is implemented across institutions, gaps resulting

from the lack of diversity in the pool of researchers who provided input may emerge and modifications may be needed to make it more universally applicable.

## Conclusion

We were able to define and document consensus across disciplines on core learning outcomes that articulate the knowledge, skills, and psychosocial attitudes, behaviors, and beliefs needed to become a researcher. The resulting comprehensive framework, the CRDF, transcends disciplines and formal training career stages. Adoption and adaptation of the CRDF will not only support individual research mentees and improvement of individual research training programs but also facilitate coordination of research workforce development across programs and training stages. The S2a and S2b Appendix includes two tools to facilitate CRDF use: 1) a tool for research training program directors to map their current program activities and assessments to identify potential gaps; and 2) a tool to complement individual development plans for mentees to use with their mentors and thesis committees in planning and mapping their development as a researcher over time.

## Future directions

Our research team is committed to developing evaluation and assessment rubrics and instruments based on the CRDF. We welcome collaborators on this work and invite those interested to contact us.

## Supporting information

**S1 Appendix.  Research Development Frameworks and Assessments included in Literature Review.**
(PDF)

**S2a Appendix.  Research Training Program Development and Mapping Tool.**
(XLSX)

**S2b Appendix.  Researcher Development Plan (RDP) Tool.**
(XLSX)

**S3 Appendix.  Code Themes, Definitions and Assignments.**
(PDF)

**S4 Appendix.  Evolution of Learning Outcomes for CRDF.**
(PDF)

**S5 Appendix.  Dissemination Contacts for National Survey.**
(PDF)

**S6 Appendix.  Open Card Sorting Results.**
(PDF)

**S7 Appendix.  Framework Mapping to CRDF.**
(XLSX)

## Acknowledgments

We thank undergraduate student Shefali Bhatt for her help with the literature search and our colleagues Drs. Amber Smith, Melissa McDaniels, Christine Pfund, and Fatima Sanchieznieto for their feedback on a preliminary draft of this manuscript.

## Author contributions

**Conceptualization:** Janet L. Branchaw, Amanda R. Butz.

**Data curation:** Amanda R. Butz.

**Formal analysis:** Janet L. Branchaw, Amanda R. Butz, Joseph C. Ayoob.

**Funding acquisition:** Janet L. Branchaw.

**Investigation:** Janet L. Branchaw, Amanda R. Butz, Joseph C. Ayoob.

**Methodology:** Janet L. Branchaw, Amanda R. Butz.

**Project administration:** Janet L. Branchaw, Amanda R. Butz.

**Resources:** Janet L. Branchaw, Amanda R. Butz.

**Supervision:** Janet L. Branchaw.

**Validation:** Janet L. Branchaw, Amanda R. Butz, Joseph C. Ayoob.

**Visualization:** Janet L. Branchaw, Amanda R. Butz, Joseph C. Ayoob.

**Writing – original draft:** Janet L. Branchaw, Amanda R. Butz.

**Writing – review & editing:** Janet L. Branchaw, Amanda R. Butz, Joseph C. Ayoob.

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
