## [Decision Letter · Decision Letter 0]

7 Aug 2025

Dear Dr. Branchaw,

Thank you for submitting your manuscript to PLOS ONE. After careful consideration, we feel that it has merit but does not fully meet PLOS ONE’s publication criteria as it currently stands. Therefore, we invite you to submit a revised version of the manuscript that addresses the points raised during the review process.

**Overall, this work generates a useful framework that can be used to support professional development of researchers. The reviewers suggested the authors clarify the results and some of the figures prior to publication.**

We look forward to receiving your revised manuscript.

Kind regards,

Amy Prunuske

Academic Editor

PLOS ONE

Journal Requirements:

**Additional Editor Comments:**

The reviewers were enthusiastic about the creation of the Comprehensive Research Development Framework, and noted several areas for improvement. I encourage the authors to address the reviewers' comments regarding the results section and Figure 4. The reviewers suggest that it might be helpful to include a flow chart and that the authors should consider being more explicit about generating a tool that demonstrated consensus across disciplines.

Reviewers' comments:

Reviewer's Responses to Questions

**Comments to the Author**

1. Is the manuscript technically sound, and do the data support the conclusions?

Reviewer #1: Partly

Reviewer #2: Yes

2. Has the statistical analysis been performed appropriately and rigorously?

Reviewer #1: N/A

Reviewer #2: Yes

3. Have the authors made all data underlying the findings in their manuscript fully available?

Reviewer #1: Yes

Reviewer #2: Yes

4. Is the manuscript presented in an intelligible fashion and written in standard English?

Reviewer #1: No

Reviewer #2: Yes

Reviewer #1: The manuscript by Branchaw, Butz, & Ayoob aims to develop a comprehensive framework to guide research training, primarily at the graduate and postdoctoral level. This is accomplished first by a literature review to establish a preliminary list of research competencies and then refined through an online survey and two rounds of card-sorting exercises by experienced researchers to organize the competencies into overall categories. The result is eight categories of research skills that contain 57 specific skills/competencies. This is an interesting study. The methodology is well thought out and the resulting framework they’ve developed has genuine applications for graduate student training across disciplines (and likely for post-docs as well). Unfortunately, the manuscript is seriously hobbled by a lack of clarity in the writing of the Results section. I’ve attempted to list specific problems below, but it might be worth the authors getting together to re-outline this section and then use that outline as a basis for a more readable Results section.

Lines 69-70: This is an important statement. Is there a citation?

It might be worthwhile for the authors to have a flow chart-like figure that explains their process of collecting competencies from the literature > surveying researchers > creating a refined list > open card sorting > creating categories > closed card sorting > refining the framework.

Regarding the card-sorting exercises. Is this an evidence-based approach? If so, what are the citations sorting the use of the open and closed sorting approaches.

It feels like the Results are missing a section about their literature review.

Lines 409-412 - Information about the 79 outcomes prepared for the survey and the 75 that actually appeared in the survey due to a survey error belongs in the Methods section. For the results section, the authors should present the data as though there were a list of 75 outcomes since that's what there is actually data on.

Figure 4 - It is not clear what figure 4 is actually supposed to show us or if it is even that important. Is this meant to communicate the % of the 75 learning outcomes judged important by the survey (segregated by discipline)? Is this really something that is important to show? What outcomes were deemed important is certainly interesting and is presented in Table 4. Figure 4's message seems to be "there were things that researchers thought were important" and that's it. Consider omitting (or explaining it better).

Table 4 - Perhaps the title should be "Learning Outcomes ... by 80% or More of Overall Respondents". Also, what is the significance of the non-bolded numbers? Just that they are below 80%?

Lines 430-432: Are the details about what categories the relatively low scored (variable importance) outcomes really needed here. It distracts from the subsequent report about the higher scoring 50 outcomes.

Lines 434-440: I'm completely confused by this section. "Some variation" is reported by discipline but not what those differences are or whether they were statistically different. Then ratings were combined to minimize those differences. If there really are differences between disciplines, then it seems like something worth talking about and maybe analyzing with distribution stats. If not, then just talk about the combined data. Also, the last sentence presents a series of numbers in parentheses. Are these the learning outcomes in Table 1? If so, the format does not correspond (there are no 3.01 or 4.04 in Table 1).

Lines 454-455: Again, why this discussion on what wasn't highly scored.

Lines 501-502: Shouldn't the authors refer to Table 5? Also, I'm not sure that Tables 5 & 6 are in the right order. Or Table 5 is confusing. The process of going to phase 1 to phase 2 to the final 8 set of categories is shown in table 5, but the process of getting to the last set of 8 categories is based on data in Table 6. Either this needs to be explained better in the text or the order of the tables should be changed.

Lines 523-526: I'm not sure I understand this approach of dealing with the learning outcomes that showed weak consensus. It sounds arbitrary, but I suspect it was not. Two potential reasons for lack of consensus were that there were discipline-specific differences in which researchers thought these outcomes should be placed. Another is that the outcome could be placed in multiple categories because it was interpreted differently by different groups. If the latter, then perhaps the right approach by the authors would be to "split" the outcome into two outcomes with each reflecting the category that researchers matched them to. An interesting follow-up experiment would be to repeat the forced choice supporting and see if the split/renamed categories got sorted as the authors predicted/assumed.

Lines 549-557: How were the Table 7 data generated. I didn't see anything in the Methods that might explain it. Also, what exactly is it saying? Is this mapping how often a given outcome was discussed in the literature reviewed? Finally, doesn't this belong in the Results section?

Line 563 paragraph: It might be worth stating that the CDRF is meant to represent a relative consensus across disciplines.

Line 576 paragraph: The use of the CRDF to increase transparency is an important point. Could the authors discuss how to operationalize that (similar to how they discussed using the CRDF as an assessment tool in the subsequent section). Placed in a student handbook? Part of a program/mentor/student compact? Something else?

This is somewhat discussed in the limitations section, but regarding the data from the Arts & Humanities. How confident are the authors regarding this data given the small sample size? Perhaps these 5 are in a post-grad research career because they had this research training in their undergrad years.

Reviewer #2: “The Comprehensive Research Development Framework (CRDF): Core Learning Outcomes for Research Training” manuscript presents an important framework that the authors developed. The process for developing the framework was excellent – supplementing the authors’ deep expertise with input from multiple stakeholders in multiple ways. The framework will be useful to a range of stakeholders (e.g., program administrators, faculty members, students, postdoctoral scholars) across a wide range of fields – including this reviewer. The manuscript is well written and easy to follow.

Below are items that could be improved in the manuscript.

-Consistency or simplification of the headings would be helpful. For example, the methods sections has STEPS 1-4, with STEP 1 having four lettered subsections; the results section has two main subsections and one of those subsections lists the two phases of card sorting; and limitations, conclusions, and future directions are all short main sections.

-Somewhere in the manuscript it would be helpful to note the geographic comprehensiveness of the framework. For example, were most of the frameworks used in the literature review from the United States (or from the US and Europe) – or was there global representation from where the frameworks were developed? Is researcher development often done similarly globally, or is this framework likely most applicable to US researcher development?

-There are some items that could be clarified on the figures and tables.

*More detail in the captions would be helpful so that the tables and figures could stand alone.

*There are some superscripts in Table 6, and the descriptions of what those superscripts represent is missing (i.e., there are not footnotes below Table 6).

*Table 7 has a column titled “framework names,” but the items listed seem to be the article titles.

*Table 1 lists the learning outcomes as 1.1, 1.2, 1.3…, and the text of the manuscript adds a “0” before the learning outcomes below 10 (e.g., 3.01 on line 439).

**Do you want your identity to be public for this peer review?** For information about this choice, including consent withdrawal, please see our Privacy Policy

Reviewer #1: No

Reviewer #2: **Yes: ** Meghann Jarchow

---

## [Author Response · Author response to Decision Letter 1]

26 Aug 2025

Editor- address the reviewers' comments regarding the results section and Figure 4

Response: See responses to individual reviewer comments below.

Editor - helpful to include a flow chart

Response: Figure 1 was revised and is now a more detailed flow chart.

Editor - consider being more explicit about generating a tool that demonstrated consensus across disciplines

Response: We added text indicating the CRDF transcends and demonstrates consensus across disciplines in multiple places in the article. See lines 32, 84, 590-1, and 654.

Reviewer #1 - the manuscript is seriously hobbled by a lack of clarity in the writing of the Results section

Response: The Methods and Results sections have been reorganized to be parallel, both following the detailed flow chart presented in figure 1. We believe this addresses the lack of clarity in the writing. The change necessitated flipping the numbering of Tables 3 & 4.

Reviewer #1 - This is an important statement. Is there a citation?

Response: Citations have been added in line 70.

Reviewer #1 - It might be worthwhile for the authors to have a flow chart-like figure that explains their process of collecting competencies from the literature > surveying researchers > creating a refined list > open card sorting > creating categories > closed card sorting > refining the framework.

Response: Figure 1 has been revised with more details as a flow chart. In addition, the Methods and Results sections have been reorganized to more explicitly reflect the process.

Reviewer #1 - Regarding the card-sorting exercises. Is this an evidence-based approach? If so, what are the citations sorting the use of the open and closed sorting approaches.

Response: The citation about use of card sorting activities has been added in line 327.

Reviewer #1 - It feels like the Results are missing a section about their literature review.

Response: A section about the literature search and review has been added to the Results in the new organization.

Reviewer #1 -Information about the 79 outcomes prepared for the survey and the 75 that actually appeared in the survey due to a survey error belongs in the Methods section. For the results section, the authors should present the data as though there were a list of 75 outcomes since that's what there is actually data on.

Response: The information about the 75 of the 79 outcomes that were included in the survey (4 were missing due to an error) has been moved from the Results to the Methods section.

The data presented in the Results now explicitly lists 75 learning outcomes in the legend of figure 4, where data collected about the importance of the learning outcomes is presented. Note the in figure 5 data from all 79 outcomes are included because all were asked about on the career stage survey questions.

Reviewer #1 - It is not clear what figure 4 is actually supposed to show us or if it is even that important. Is this meant to communicate the % of the 75 learning outcomes judged important by the survey (segregated by discipline)? Is this really something that is important to show? What outcomes were deemed important is certainly interesting and is presented in Table 4. Figure 4's message seems to be "there were things that researchers thought were important" and that's it. Consider omitting (or explaining it better).

Response: Figure 4 provides evidence of content validity from experts for the core learning outcomes. A statement explicitly explaining this has been added in lines 417 – 418.

Reviewer #1 - Perhaps the title should be "Learning Outcomes ... by 80% or More of Overall Respondents". Also, what is the significance of the non-bolded numbers? Just that they are below 80%?

Response: Table 4 was simplified to report only the overall importance ratings across disciplines and the bolded numbers were removed.

Reviewer #1 - Are the details about what categories the relatively low scored (variable importance) outcomes really needed here. It distracts from the subsequent report about the higher scoring 50 outcomes. I'm completely confused by this section. "Some variation" is reported by discipline but not what those differences are or whether they were statistically different. Then ratings were combined to minimize those differences. If there really are differences between disciplines, then it seems like something worth talking about and maybe analyzing with distribution stats. If not, then just talk about the combined data. Also, the last sentence presents a series of numbers in parentheses. Are these the learning outcomes in Table 1? If so, the format does not correspond (there are no 3.01 or 4.04 in Table 1). Again, why this discussion on what wasn't highly scored.

Response: We agree with the reviewers’ comments and have followed their recommendation to remove the details about possible small disciplinary differences and to focus on the main findings.

Reviewer #1 - Shouldn't the authors refer to Table 5? Also, I'm not sure that Tables 5 & 6 are in the right order. Or Table 5 is confusing. The process of going to phase 1 to phase 2 to the final 8 set of categories is shown in table 5, but the process of getting to the last set of 8 categories is based on data in Table 6. Either this needs to be explained better in the text or the order of the tables should be changed.

Response: Table 5 precedes table 6 because table 5 shows the results of the first (open) card sort and table 6 the results of the second (closed) card sort. Table 5 also shows the final categories, which were finalized based on the second (closed) card sort results. To clarify this, the table 5 column headers were modified, and the text has been modified to more explicitly describe what is in each table using “open” and “closed” card sorting (rather than first and second phase).

Reviewer #1 - I'm not sure I understand this approach of dealing with the learning outcomes that showed weak consensus. It sounds arbitrary, but I suspect it was not. Two potential reasons for lack of consensus were that there were discipline-specific differences in which researchers thought these outcomes should be placed. Another is that the outcome could be placed in multiple categories because it was interpreted differently by different groups. If the latter, then perhaps the right approach by the authors would be to "split" the outcome into two outcomes with each reflecting the category that researchers matched them to. An interesting follow-up experiment would be to repeat the forced choice supporting and see if the split/renamed categories got sorted as the authors predicted/assumed.

Response: The explanation for how we dealt with the learning outcomes for which there was low consensus in the closed card sorting exercise has been clarified in lines 534 – 539 by adding more detail about the decision-making process.

Reviewer #1 - How were the Table 7 data generated. I didn't see anything in the Methods that might explain it. Also, what exactly is it saying? Is this mapping how often a given outcome was discussed in the literature reviewed? Finally, doesn't this belong in the Results section?

Response: A section was added to the Methods “Back Map Source Framework Elements to Learning Outcomes and Categories to Confirm Coverage” to describe this process (lines 360 – 365), and Table 7 was moved from the Discussion to the Results section.

Reviewer #1 - It might be worth stating that the CDRF is meant to represent a relative consensus across disciplines.

Response: Great suggestion! We added text indicating the CRDF transcends and demonstrates consensus across disciplines in multiple places in the article. See lines 32, 84, 590-1, and 654.

Reviewer #1 - The use of the CRDF to increase transparency is an important point. Could the authors discuss how to operationalize that (similar to how they discussed using the CRDF as an assessment tool in the subsequent section). Placed in a student handbook? Part of a program/mentor/student compact? Something else?

Response: Specific examples of how the CRDF can be used for and with research mentees have been added in lines 617 – 620.

Reviewer #1 - This is somewhat discussed in the limitations section, but regarding the data from the Arts & Humanities. How confident are the authors regarding this data given the small sample size? Perhaps these 5 are in a post-grad research career because they had this research training in their undergrad years.

Response: We are confident in the Arts & Humanities data and have therefore included it, but because of the small sample size we are reluctant to draw sweeping conclusions from it. We believe that acknowledging and discussing it in the Limitations section is the best way to convey this and have elaborated on what is written there to be more explicit. See lines 644 – 645.

Reviewer #2 - Consistency or simplification of the headings would be helpful. For example, the methods sections has STEPS 1-4, with STEP 1 having four lettered subsections; the results section has two main subsections and one of those subsections lists the two phases of card sorting; and limitations, conclusions, and future directions are all short main sections.

Response: Figure 1 has been revised with more details as a flow chart outlining the process we used and headings (rather than numbers and letters) have been used in the revised Methods and Results sections.

Reviewer #2 - Somewhere in the manuscript it would be helpful to note the geographic comprehensiveness of the framework. For example, were most of the frameworks used in the literature review from the United States (or from the US and Europe) – or was there global representation from where the frameworks were developed? Is researcher development often done similarly globally, or is this framework likely most applicable to US researcher development?

Response: 34% (19) of the source frameworks were developed outside the United States, including: Iran, Germany (2), United Kingdom (3), South Africa (2), Turkey, Canada (2), Peru, Australia (3), Netherlands, India, China, and Hong Kong. A statement about this was added to the Results in lines 375 - 376 and in the discussion in lines 570 – 573.

Reviewer #2 - More detail in the captions would be helpful so that the tables and figures could stand alone.

Response: We reviewed the captions and feel confident that they convey all the information needed to interpret the tables and figures. It is unclear what the reviewer means by “stand alone.” We do not think it necessary to replicate the text from the Methods and Results sections in the captions.

Reviewer #2 - There are some superscripts in Table 6, and the descriptions of what those superscripts represent is missing (i.e., there are not footnotes below Table 6).

Response: The superscript notations are now referenced at the bottom of the table.

Reviewer #2 - Table 7 has a column titled “framework names,” but the items listed seem to be the article titles.

Response: The column title has been changed to “Framework Article.”

Reviewer #2 - Table 1 lists the learning outcomes as 1.1, 1.2, 1.3…, and the text of the manuscript adds a “0” before the learning outcomes below 10 (e.g., 3.01 on line 439).

Response: The learning outcome numbers have been updated to include the 0 in those below 10 (e.g., 1.1 is now 1.01).

---

## [Editor Report · Decision Letter 1]

2 Sep 2025

The Comprehensive Researcher Development Framework (CRDF): Core Learning Outcomes for Research Training

PONE-D-25-37622R1

Dear Dr. Branchaw,

We’re pleased to inform you that your manuscript has been judged scientifically suitable for publication and will be formally accepted for publication once it meets all outstanding technical requirements.

Kind regards,

Amy Prunuske

Academic Editor

PLOS ONE
---

## [Editor Report · Acceptance letter]

PONE-D-25-37622R1

PLOS ONE

Dear Dr. Branchaw,

I'm pleased to inform you that your manuscript has been deemed suitable for publication in PLOS ONE. Congratulations! Your manuscript is now being handed over to our production team.

Kind regards,

on behalf of

Dr. Amy Prunuske

Academic Editor

PLOS ONE